# Experiences and effect of implementing social health insurance (SHI) program in Nepal-A mixed method study

**Rajani Bharati**[1]*, **Diana Romero**[1], **Alexis Pozen**[1], **James Sherry**[1], **Bhuwan Paudel**[2], **Mukesh Adhikari**[3], **Prakash Acharya**[4]

1 CUNY Graduate School of Public Health and Health Policy, New York, New York, United States of America, 2 Ministry of Health, Kathmandu, Bagmati, Nepal, 3 Gillings School of Global Public Health, University of North Carolina, Chapel Hill, North Carolina, United States of America, 4 Missouri Heart Center, Columbia, Missouri, United States of America

* rbharati.ach@gmail.com

## Abstract

Nepal initiated the National Health Insurance Program (NHIP) in 2016, but enrollment rates remain low, with an increasing dropout rate. This study examined the experience of service providers and beneficiaries with NHIP and its effect on insurance enrollment and health service utilization. We employed an exploratory sequential mixed-method design, involving 14 focus group discussions and 20 key informant interviews, supplemented by quantitative data from Nepal's District Health Information System (DHIS-2). We identified a complex interconnection between demand- and supply-side factors affecting enrollment, renewal, and health-seeking behavior within NHIP. Both NHIP enrollees and non-enrollees cited the quality of health services as crucial for enrollment. Other significant factors included inadequate awareness, insurance scheme design (service lag time, referral requirements), limited geographical accessibility to health facilities, ability to pay premiums, perceived illness risk, and the perceived usefulness of health insurance. Service providers reported that NHIP implementation increased patient flow and administrative burden without proportional resource growth. They faced challenges such as a lack of motivation, claim and reimbursement difficulties, tedious medicine procurement, and insufficient information about the insurance program. Our quantitative analysis confirmed increased patient flow and referrals due to the policy's effect. In intervention districts, there was an increase in the average number of total client visits, new client visits, and referrals by 4,729, 2,721, and 163 respectively, compared to comparator districts. These increases occurred when the enrollment rate was at 5%. Our findings highlight the dependency of NHIP enrollment on the quality of the healthcare delivery system. To improve NHIP effectiveness, there should be increased awareness and insurance literacy, enhanced insurance scheme design features, and improved geographical accessibility to health facilities. Efforts should also focus on resource availability, expanding the qualified health workforce, and improving stewardship and accountability mechanisms.

**Data availability statement:** The data supporting this study are available in Figshare. The DHIS-2 dataset can be accessed at https://doi.org/10.6084/m9.figshare.25715556.v1, and the Post Focus Group Responses are available at https://doi.org/10.6084/m9.figshare.25715559.v1.

**Funding:** The author(s) received no specific funding for this work.

**Competing interests:** The authors have declared that no competing interests exist.

## Introduction

More than half of the world's population does not receive all the essential services they need [1]. In 2019, over a billion people worldwide spent at least 10% of their household budget on health care and approximately 344 million people were pushed into extreme poverty because of unexpected out-of-pocket (OOP) health expenses [1]. In May 2005, the World Health Assembly passed a policy resolution for the World Health Organization (WHO), whereby WHO considered social health insurance (SHI) as a promising means to achieve universal health coverage (UHC) [2].

SHI pools risks and mobilize the funds to finance health services, allowing enrollees to contribute based on their ability to pay and ensuing that all contributors receive pre-defined benefits, regardless of income or social status [3,4]. There are multiple variations of SHI from universal and mandatory membership to voluntary membership for those in the informal sector, often subsidized by the government [3,4]. The move towards UHC is typically incremental, with the transition period varying by country [5]. Global experiences highlight the importance of factors such as economic growth, population distribution, SHI administration capabilities, societal solidarity, and governmental stewardship [5]. Nevertheless, many low- and middle-income countries are adopting SHI to achieve UHC [6].

In Nepal, diverse healthcare financing strategies have historically been implemented, including user fees in public facilities, free healthcare policies, community-based insurance schemes, and several non-governmental community initiatives [2]. However, these interventions failed to provide adequate financial protection or meet the medical needs of target groups [2]. Nepal introduced its National Health Insurance Program (NHIP) under the National Health Insurance Policy in 2013 [7], driven by rising OOP spending (from 43% in 2010 to 60% in 2017) [8–10] and a commitment to achieve UHC by 2030 [7]. The NHIP aims to ensure equitable access to quality health services, financial protection against catastrophic illness, strengthen the health system, and improve the health-seeking behavior of the public [7].

Initially launched in three pilot districts in 2016 (Kailali, Baglung, and Illam), the NHIP went through a staggered expansion and has reached 75 districts [11]. The key features include an autonomous governing body, single risk pool, gatekeeping system in the service delivery, family-based membership, DRG payment system to providers, and subsidization of premiums to certain vulnerable groups [12] (Table 1). Public financing alone is not feasible in Nepal due to large informal sector and lack of means to assess income. Therefore, NHIP's revenue primarily comes from premiums and complementary government funding at subsidies for exempt categories [12]. The Health Insurance Act envisions universal enrollment, but it remains voluntary, and global experiences show the transition to mandatory enrollment can be slow. Initially, the NHIP enrollment averaged around five percent in the first two years [13–16]. Policy changes, such as subsidies for seniors and vulnerable groups (Table 1),increased enrollment to approximately 17% in fiscal year (FY) 2019/20 [13–16]. Notably, at least one-third of the enrollees were subsidized [17]. Dropout rates also emerged as a significant concern, with national rates of 44.5% and 38.4% in 2017/18 and 2018/19, respectively [15,16]. In Kailali district alone, dropout rates reached 77% in FY 2017/18 [15].

With an increased enrollment among high-risk groups and a high dropout rate among presumably healthy populations, understanding the determinants of insurance enrollment becomes crucial. Additionally, some tertiary-level public hospitals have not participated, foreboding service providers' issues. Few studies have explored the NHIP's demand-side determinants [18,19] or has assessed its impact on healthcare utilization [20]. This study aims to gain a holistic insight from both the demand and supply side, exploring perceptions

**Table 1. Nepal's national health insurance program design features.**

| NHIP design features [12] | Details |
|---|---|
| Governing body | Health Insurance Board, an autonomous body that function independently of the Ministry of Health; serves as a purchaser and implements the health insurance program. |
| Legal Framework | The Constitution of Nepal 2015 endorses the right to seek basic health services and equal access to health care [21] Health Insurance Act 2017 Endorsement of the Health Insurance Regulation, 2019 |
| Population coverage | Voluntary enrollment at present. The Health Insurance Act of 2074, enacted in October 2017, aims for universal enrollment in health insurance. The Act includes provisions for enrolling various groups such as national service guards, migrant workers, residents of elderly homes, orphanages, correctional facilities, protection centers, and employees of "prathisthan." A "prathisthan" refers to any profit-making or non-profit-making company, private firm, partnership firm, cooperative, or similar entity. These measures were introduced to cover both the formal sector and individuals who might otherwise be overlooked. However, these additional provisions have not yet been implemented. |
| Financing and membership | Family-based membership and premium contribution Employee contribution (yet to be started) Complementary funding from the government's budget |
| Contributions | NPR 3500 (USD 3.5) per annum for HH of 5 members NPR 700 per additional member Salary contribution: 1% of base salary for each employee and employer (yet to be started) Subsidy for certain categories |
| Subsidies on contribution amounts | 100% subsidy- families with ultra-poor identification cards 100% subsidy- vulnerable groups including families with members who have complete disability, leprosy, HIV, and multidrug-resistant TB; 100% subsidy- senior citizen (70 years and above) 50% subsidy- Female Community Health Volunteers (FCHVs) |
| Registration and Service start date | Four fixed service start dates in a year. There is a lag time of one to four months between registration and service start date. For instance, those registered or renewed between January 16 and April 15 will have a service start date of May 15. |
| Risk pooling and risk protection | Single fund with no fragmentation |
| Benefit package | Health promotion, prevention, treatment, rehabilitation, and ambulance service package with a benefit ceiling of NPR 100,000 for a family of five per year. An additional benefit of NPR 20,000 for each added member not exceeding a maximum of NPR 200,000 per family. Senior citizens get a separate benefit amount of up to NPR 100,000. Ambulance service only covers emergency conditions |
| Health care delivery | Both public and private health facilities with gatekeeping Only public facilities are allowed to serve as a "Primary service point". Enrollees choose their primary service point during registration. Any facilities that are not primary service points can provide only emergency and referral services. Referral cards are only valid for seven days. HIB enters into contractual agreements with health facilities, which is valid for five years. |
| Payment to providers | Capitation, per diem, and case-based (Diagnosis Related Groups) HIB pays providers based on various packages and individual items. The packages include Diagnosis based, per day basis, OPD, Emergency, Eye hospital outpatient/emergency, cardiac services, Surgical, and others. The same payment rate for public and private facilities HIB does not pay if health facilities are providing services under other vertical programs such as free health programs, safe motherhood programs, and family planning programs. |
| Remuneration to Enrollment Assistants (EAs) | 10 to 15% of the premium amount depending on the performance. |
| Claim management and Information System | Integrated Management Information System (IMIS) |

of enrollees and non-enrollees, challenges faced by service providers, and NHIP's impact on health service utilization. Specific aims include exploring the demand side (enrollees and non-enrollees) determinants affecting insurance enrollment; understanding the service providers' experiences and challenges in providing health services under the NHIP; and assessing NHIP's effect on health service utilization in terms of 'total client visits', 'new clients visits', and referral outs.

## Materials and methods

### Ethics statement

Ethical approval was obtained from the Institutional Review Board (IRB) of the City University of New York (CUNY) (protocol # 2019–0537) and the Nepal Health Research Council (review # 568/2019). We explained the study objectives, benefits, risks, right to refuse participation, and confidentiality of responses to all the participants. Both verbal and written informed consent were obtained from the participants.

**Study design.**  The study utilized an exploratory sequential mixed-method design [22,23]. Initially, qualitative data were collected which revealed an increase in patient visits and referrals at health facilities. Based on this qualitative information, the quantitative portion was designed to examine healthcare utilization in terms of total client visits, new client visits, and referral outs. The findings from both qualitative and quantitative analysis were then combined and displayed using the fishbone diagram. Focus group discussions and Key Informant (KI) interviews were used to explore the perceptions and experiences of insurance enrollees, non-enrollees, and service providers. The difference-in-difference (DID) approach was used to assess the NHIP's impact on health service utilization.

**Study setting, population, sampling, and recruitment.**  The qualitative data was collected from two NHIP pilot districts- Baglung and Kailali, ensuring representation of urban and rural settings. KIs included Health Insurance (HI) focal persons, physicians, nurses, medical administrators, Enrollment Officers (EOs), and Enrollment Assistants (EAs) from PHCCs, district/regional government hospitals, private hospitals, and province/district health insurance offices. The focus group participants included health insurance enrollees, ex-enrollees, and non-enrollees. Ex-enrollees, who had experience with the program but did not renew their membership, were grouped with enrollees for discussions. Anyone who could engage in constructive discussions about NHIP was purposively selected for the study.

DHIS-2 data [24] from all the health facilities in 73 out of 77 districts of Nepal were used for the quantitative section of this study. Four districts were excluded as they started insurance service late in FY 2017/18. The 20 districts that started the insurance program in 2016 and 2017 were regarded as intervention districts; 53 districts that had not started the program or started after May 2018 were categorized as non-intervention districts. (S1 Table). Since only public health facilities are mandated to report on DHIS-2, the data do not include medical colleges, private hospitals, and tertiary centers. In the final analysis, we used 240 health facilities in FY 2014/15 (pre-intervention) and 251 facilities in FY 2017/18 (post-intervention) [25] (Fig 1).

**Data collection and analysis.**  The data was collected between 25/11/2019–10/01/2020. At the time of data collection, only a few studies had explored the demand and supply side challenges in implementing NHIP. Systematic reviews on the impact of SHI in low- and middle-income countries have shown mixed results regarding health service utilization [26,27]. Therefore, we used the grounded theory approach for the data collection and analysis [28]. Twenty three key informant interviews (87% response rate), 14 focus group discussions (six in Baglung and eight in Kailali district), and post-focus group surveys with focus group participants were administered using the semi-structured interview guides and questionnaires (S1 Text). The information obtained during data collection was used to reformulate the recruitment and data collection strategies. The information was also used to probe the discussions during interviews and focus groups. All the instruments were translated into the Nepali language. In-person interviews were mainly held in the office setting of health facilities, province/district HI offices, or at EAs house. Focus groups were conducted at health facilities, EA's or community social workers' houses, and school/temple yards.

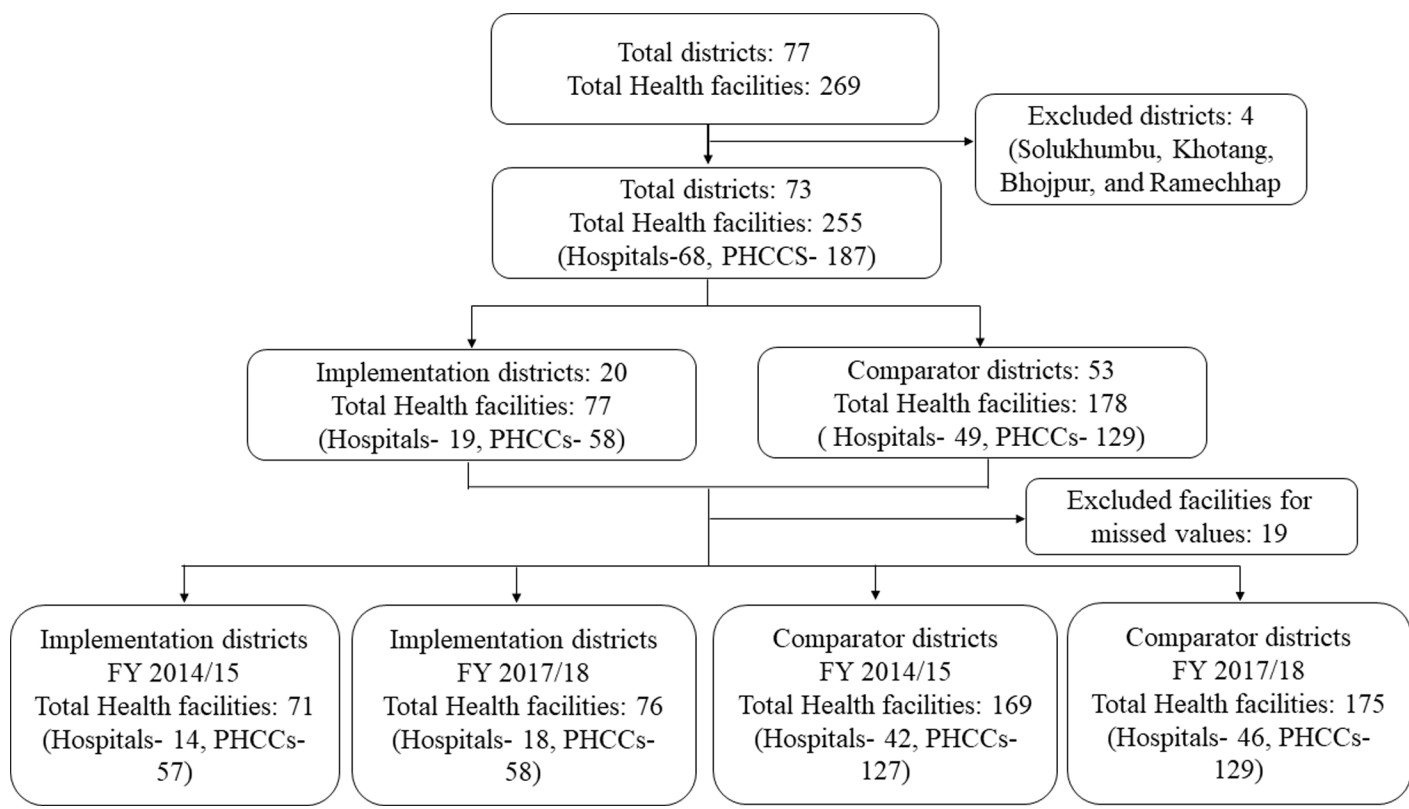

**Fig 1. Flowchart showing the selection of districts and health facilities.**

Each interview and focus group were audiotaped, transcribed, and translated into English. Two coders thoroughly reviewed the transcripts. The first coder reviewed all the transcripts and set the initial codes, while the second coder reviewed 25% of the transcripts (representative of all subgroups) and assigned the developed codes to the transcripts. The initial inter-coder reliability for interview data was 97%, and the focus group was 96%, which was increased to 100% by sorting, synthesizing, and integrating the codes into sub-themes and themes. The coding process continued until new interview/focus group data did not alter the definition or score of the codes. Dedoose software was used for the coding, memoing, and analytical functionality [29]. After the focus group session, post-focus group survey was conducted with participants. The survey data [30] was then analyzed using Microsoft Excel to compare the demographic characteristics of enrollees and non-enrollees.

The quantitative data was obtained from the DHIS-2 (Nepal's national health information management system) through the Integrated Health Information Management Section under the Management Division, Department of Health Service (DHS) Nepal. Regarding the analysis, trends of the mean values of outcome variables were examined for four consecutive years, starting from FY 2014/15 to FY 2017/18, to visually inspect the parallel trend. Once the assumption of the parallel trend for Difference in Difference (DID) was met [31,32] (S1 Fig), the DID estimation was made to calculate the average treatment effect of the NHIP comparing implementation and comparator districts between FY 2014/15 and FY 2017/18.

We used the DID model [33]:

$$Y_{hgt} = \alpha + \beta\,Post_t + \lambda\,Treatment_g + \delta\left(Post_t * Treatment_g\right) + \varepsilon$$

Here, g indexes the group (g = 0 for the comparison district, 1 for the implementation district), and t represents time (0 = 2014/15, 1 = 2017/18).

Y is the outcome of interest for time t and health facility h in group g. Alpha (α) is the intercept. Beta (β) represents the parameter for the time component (pre- or post-intervention). Lambda (λ) represents the parameter for the group fixed effects. Delta (δ) *represents the interaction between time and group effect, which is the parameter of interest and is called the difference-in-difference estimator. It estimates the average treatment effect of the policy (the effect of treatment on the average outcome of Y) for districts that implemented the health insurance program compared to those that did not.* Epsilon (ε) is the error term.

The outcome variables included total client visits, new client visits, and referral outs. We also conducted a subgroup analysis based on the types of health facilities (hospitals and PHCCs). The HIB aimed to balance the number of implemented districts among the provinces while conducting staggered implementation. Therefore, we did not include any time-varying district-level control, assuming that the districts included in the intervention groups closely resemble the districts in the control group. Additionally, we did not incorporate facility-level covariates into the model, such as the availability of resources, due to a large number of missing data in DHIS-2.

We removed the 19 facilities with missing values for outcome variables from the analysis (15 in pre-intervention and four in post-intervention years). The facilities with missing values were spread across the country and in both intervention and non-intervention districts. None of the districts had more than one facility with missing data.

All the statistical analyses for quantitative data were conducted using STATA 14.0 software [34].

## Results

A total of 97 individuals participated in 14 focus group discussions, of which 49 were insurance enrollees and the remaining 48 were non-enrollees. Nearly all participants also completed the post-focus group survey questions. Most focus group participants were female and had an education below high school. The mean age of participants was 45 years for enrollees and 37 years for non-enrollees. The proportion of respondents who were female, Dalits (a historically marginalized ethnic group in Nepal), had higher annual income, and used private health facilities explicitly were higher among the non-enrollee group. In contrast, the proportion of those with chronic illness in the family or a history of catastrophic illness in the family was higher among the enrollee group (Table 2).

Among the 23 key informants, most (n = 17) were males. The largest percentage of informants were affiliated with hospitals (55%), followed by PHCCs (25%) and HI province/district offices (20%). Various themes emerged from the focus group. The quality of health services was a central factor influencing enrollment and renewal to the NHIP, as well as overall health-seeking behavior. Other significant factors included lack of awareness, insurance scheme design features (such as service lag time, requirement of referral, and validity of referral card for seven days), geographical accessibility to health facilities, ability to pay premiums, perceived risk of getting and illness and perceived usefulness of health insurance programs. Likewise, service providers or key informants highlighted several challenges in offering health services under NHIP. A major challenge was the increased patient flow and administrative burden without a proportionate increase in resources. Other challenges included the motivation of health service providers and enrollment assistants, difficulties in making claims and reimbursements, tedious medicine procurement processes, and insufficient information about the insurance program.

Table 2. Demographic and health-related characteristics of focus group participants.

| Characteristics | Insurance enrollees | Insurance non-enrollees |
|---|---|---|
| Districts | N = 49 | N = 48 |
| Baglung | 23 (47%) | 19 (40%) |
| Kailali | 26 (53%) | 29 (60%) |
| Age (in years) | N = 49 | N = 48 |
| (mean/SD) | 45 yrs (12) | 37 yrs (13) |
| Sex (N/%) | N = 49 | N = 48 |
| Male | 12 (25%) | 3 (6%) |
| Female | 37 (75%) | 45 (94%) |
| Caste/ethnicity (N/%) | N = 49 | N = 48 |
| Brahman/Chhetri | 29 (59%) | 22 (46%) |
| Janajati | 12 (25%) | 10 (21%) |
| Dalit | 8 (16%) | 16 (33%) |
| Head of Household (N/ %) | 27 (55%) | 16 (33%) |
| Education | N = 49 | N = 48 |
| Never attended school | 20 (41%) | 14 (29%) |
| Primary School (up to grade 5) | 12 (24.5%) | 14 (29%) |
| Secondary School (up to grade 10) | 12 (24.5%) | 10 (21%) |
| High School or +2 | 4 (8%) | 9 (19%) |
| Bachelor's degree | 0 | 1 (2%) |
| Master's degree or higher | 1 (2%) | 0 |
| Annual income (in NRP) | N = 32 | N = 36 |
| (mean/SD) | 257,383 (173,339) | 221,666 (397,179) |
| Family size (number of members) | N = 49 | N = 48 |
| (mean/SD) | 6 (3) | 5 (2) |
| Chronic illness in family members (N/%) | 30 (61%) | 20 (42%) |
| History of catastrophic illness in the family (N/%) | 27 (55%) | 19 (40%) |
| Enrollment of all family members in insurance (among enrollees) | 35 (73%) | NA |
| The average age of the youngest family member that is not enrolled in the insurance program (among enrollees) | 11 years (8) | NA |
| Satisfaction with the insurance program | N = 48 | NA |
| Very Satisfied | 7 (15%) | |
| Somewhat satisfied | 33 (69%) | |
| Not satisfied at all | 8 (16%) | |
| Facilities where they primarily obtain health services | N = 49 | N = 48 |
| Government facilities only | 37 (76%) | 27 (56%) |
| Private facilities only | 2 (4%) | 11 (23%) |
| Both government and private facilities | 10 (20%) | 10 (21%) |
| Type of services used under the insurance program | N = 46 | NA |
| Outpatient care only | 23 (50%) | |
| Other services such as inpatient, emergency, and specialist care, including outpatient care | 23 (50%) | |
| Satisfaction with primary health centers under the Insurance program | N = 46 | NA |

(Continued)

**Table 2.** (Continued)

| Characteristics | Insurance enrollees | Insurance non-enrollees |
|---|---|---|
| Very satisfied | 5 (11%) | |
| Somewhat satisfied | 30 (65%) | |
| Not satisfied at all | 11 (24%) | |

Rather than being discrete, the determinants described by enrollees and non-enrollees and the challenges explained by service providers were found to be overlapping and intricately connected. The interaction between these factors varied, with some being mutually dependent and others being consequences of each other. We have tried to capture this complex relationship using a Fishbone diagram (Fig 2). The details are described below.

**Quality of health services** played a central role in the decision to enroll in the NHIP. Majority of the focus group participants considered the availability of resources (medicine, skilled workforce, diagnostic services), respectful service providers, short wait times, and clean, easily navigable health facilities as essential elements of quality health services. The quality of health services in public facilities was already a concern. The increased patient load and administrative burden, without concomitant increase in resources, worsened the situation.

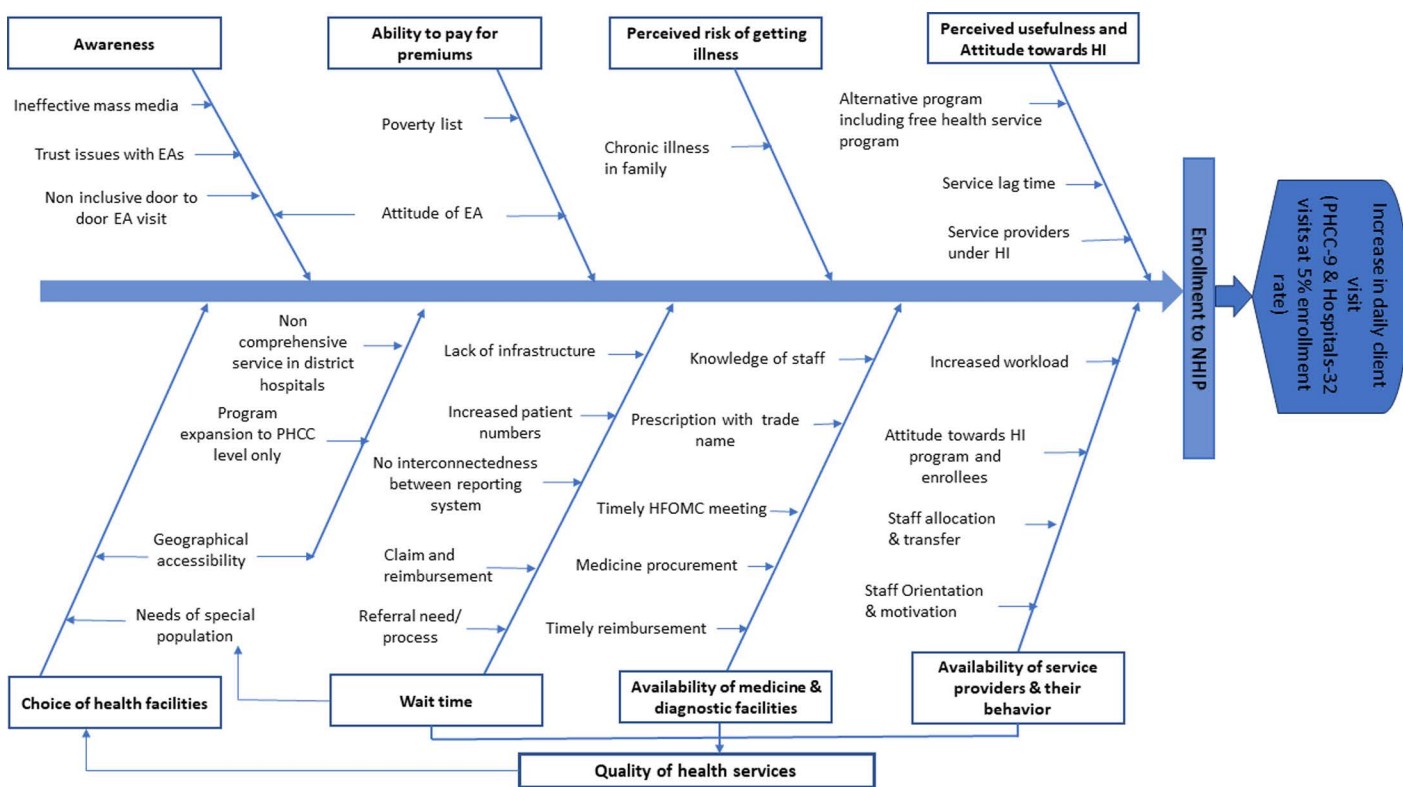

**Fig 2. Fishbone diagram showing the relationship between various demand and supply side factors in Health Insurance enrollment and health service utilization.**

**Availability of medicine and diagnostic facilities.** The unavailability of medicine was the primary cause of discontentment among insurance enrollees, leading to non-renewal. A participant mentioned "*I have decided not to renew my insurance anymore. My husband spent 20-25,000 last time on medicine even when we have insurance. If we have to buy our medicine from outside, what is the point of having insurance?*" *(Enrollee, FG 3)* Key informants outlined various reasons for the unavailability of medicine, such as tedious procurement system (a requirement of NHIP), delays in reimbursement from HIB, difficulties in timely convening Health Facility Operation and Management Committee (HFOMC) meetings, frequent absence of medical officers required for purchase approval, and shortage of essential medicines under free health services (which are not covered by HIB).

Many enrollees reported disparities in medicine provision as one stated, "*They provide different medicine for non-enrollees and cheap, low-quality medicine for us. Even if medicine is available, they withhold it once we show our insurance card.*" (Enrollee, FG 13). Several providers acknowledged maintaining different medicine sets and often offering the cheapest options to enrollees due to bidding processes favoring low-cost contracts under NHIP. Additionally, HIB encourages using generic medicine and has a fixed payment rate. Facilities may prioritize cheaper medicine procurement to maximize profits. Mistrust with NHIP medicines further escalated due to discrepancies between prescribed brand names and dispensed medicines. Some physicians were unaware of the need to prescribe generic names, while others hesitated to use generics, citing perceived differences in efficacy depending on the brand.

The lack of diagnostic facilities significantly influenced referrals, as people were reluctant to spend time navigating multiple health facilities for appropriate treatment. While some facilities reported acquiring diagnostic equipment after enrolling in the NHIP program, there were complaints about insufficient staffing and a lack of skilled personnel to utilize these resources effectively.

**Availability of service providers and their behavior.** Many focus group participants mentioned the unavailability of skilled health workers and their unpleasant behavior as their prime reason for avoiding public health facilities. One participant expressed frustration, stating, "*We rarely see doctors, and we are not even talking about specialties… we never receive prior notice about their leave. We wait in line for hours and then find out the doctor is on leave*" (Non-enrollee, FG 4).

Service providers in public facilities expressed concerns about insufficient staffing and infrastructure to match the increased patient flow and administrative workload after implementing NHIP. PHCCs, for instance, typically have only one sanctioned doctor's position, which often remains unfilled. Some PHCCs with robust HFOMCs have lobbied Village Councils to hire new staff and provide training to existing staff. However, the high staff turnover has resulted in frequent training needs, often resulting in the prolonged absence of staff from health facilities. Furthermore, there is a lack of additional sanctioned positions like pharmacist or administrative assistants necessary to meet NHIP requirements such as medicine procurement processes, and IMIS reporting.

Some front-line service providers expressed their frustration in dealing with beneficiaries' misconceptions and expectations. A few reported facing numerous "unreasonable" demands, including requests for fruit juices, unnecessary referrals to higher centers, and transportation costs from insurance enrollees. One provider voiced their concern, saying, "*We are already overwhelmed by increased patient flow and do not have time to address the misunderstandings. When people start complaining and arguing, it becomes very difficult to maintain a calm demeanor*" (Physician, 1).

**Wait time.** Participants reported long wait times as a significant deterrent to choosing public health facilities. Wait times were already an issue in public health facilities, and they

worsened with an increased patient flow and constricted time for patient care caused by administrative burdens.

The NHIP's requirement for establishing a primary contact point and need for referral has amplified the patient flow in public health facilities. Furthermore, the referral card remains valid for only 7 days, necessitating patients to obtain a new referral for follow-up visits after being discharged from referral centers. Patients must be physically present to receive a referral, as service providers claim they cannot process claims without examining patients. One participant shared, *"It took me over seven days to arrange and secure an appointment at the tertiary health center and my referral card expired by then. Since I had to return to village to obtain another referral card, which I couldn't manage, I couldn't use my insurance card"* (Enrollee, FG 3). Although the patient flow has significantly increased in public facilities, particularly PHCCs, the number of individuals actually receiving services remains relatively low. As one informant explained, "*Lately we don't have doctors. So, we mostly have referral cases. We refer 15-20 patients in a day*" (PHCC In charge- 3). Additionally, patients often request referrals specifically for private facilities, as private facilities can only take referrals or emergency cases.

Many providers reported increased administrative burden due to the new IMIS reporting requirement and lack of connectivity with other systems like DHIS-2. Some found the claim system confusing, especially for claiming multiple diagnoses and unforeseen complications.

Private facilities did not face issues with resource availability or wait times but they reported charging insurance enrollees the difference (gap amount) between NHIP coverage and their rate. A representative from a private hospital stated, "*The rate provided by HIB is too low. We either offer inadequate care or face financial difficulties. Providing subpar care would harm our reputation, so we charge patients to cover the cost difference.*" (HI focal person, private facility, 1). Unfortunately, this situation has led to misunderstandings about NHIP's promise of no copayments.

**Choice of health facilities** depended primarily on the perceived health service quality, geographic accessibility, and facilities' ability to cater to special populations like children, the elderly, and individuals with disabilities. Many people opt for private facilities or seek treatment in neighboring Indian facilities, especially in border districts. This preference reduces the demand for NHIP since only select private facilities accept insurance, and coverage is limited to emergencies and referrals. Several other factors also influence this choice including proximity to primary service points, the higher opportunity cost of seeking treatment at government facilities with long wait times, and service provider behavior.

**Lack of awareness** about NHIP was another significant factor influencing enrollment. Despite efforts through mass media, many non-enrollees were unaware of the program's existence. Some EAs with social standing were motivated to conduct door-to-door visits. However, others primarily engaged with individuals who contacted them directly for enrollment, avoiding efforts to reach the uneducated, lower-income, and hard-to-reach population due to the time-consuming nature of persuasion and low compensation. One EA mentioned, *"We have to spend atleast about 5 days to get one family enrolled and get paid only NPR 300 for five days' worth of work."* (EA 1).

Moreover, there was a lack of trust in EAs, with many perceiving them as mere "*insurance agents*." Healthcare providers, usually trusted sources, lacked sufficient knowledge and time to provide information. The absence of insurance registration services at health facilities further compounded the issue.

For existing insurance enrollees, health insurance literacy played a significant role in the renewal process. The lack of comprehensive information resulted in misunderstandings, leading to discontentment with the program. One participant expressed disappointment,

stating, "*When enrolling us, they assured that everything would be covered. But the health facility denied reimbursements. So, I did not renew the insurance anymore.*" (Enrollee, FG 9). This discontentment not only impacted renewal rates but also triggered negative word-of-mouth, dissuading potential new enrollees as one participant remarked "*No one has shared a positive experience. Even the insured people do not recommend that we enroll*" (Non-enrollee- FG 4).

**Ability to pay for premiums.** Despite subsidies for the poor, low-income individuals had low enrollment due to ineffective means-testing. NHIP relied on the poverty list created through a poor identification survey in 2015. Identified poor families were provided with a red card and EAs were supposed to use those red cards as verification to enroll the people into NHIP free of costs. However, participants reported that the poverty list did not accurately identify the real poor people and emphasized that EAs had not reached out to poor people. Conversely, EOs and EAs expressed apprehension about using the poverty list, as one explained, "*I witnessed a community dispute arising from the poverty list and I don't want to entangle with that.*" (Enrollment Officer-1).

The **perceived risk of getting ill** played a significant role in enrollment. Primarily, individuals with chronic illnesses and those requiring regular medications were inclined to enroll in the program. However, enrollment was often dependent on the availability of medicine at the primary service center. Families with more than five members typically refrained from enrolling those who were less likely to get sick, usually the youngest members. Interestingly, the possibility of catastrophic illness or accidents did not motivate people to enroll in the program, likely due to the reported poor service quality of public health facilities as one participant elucidated, "*Although I am insured, I might not use it during acute events like accidents or emergencies due to the significant delays in receiving service at government facilities*" (Enrollees, FG 12).

**Perceived usefulness of health insurance** varied among KIs, insurance enrollees, and non-enrollees. While some KIs appreciated the concept, many doubted its effectiveness in reaching vulnerable populations and the health systems' readiness. A respondent shared, "*The health institutions have not taken it as an opportunity but as a liability. If the health system functions properly, insurance is a nice program. However, our health facilities are not up to mark*" (HI focal person, 8).

Many enrollees and non-enrollees often did not perceive health insurance as useful considering it unnecessary for minor illnesses, especially with the availability of concurrent free health service programs and alternative financing mechanisms (public service employees and uniformed service employees, private-sector employees). Some participants see it as a waste of money if they do not spend enough on treatment to match their premium. The absence of major tertiary referral centers and private hospitals as service providers further diminished the program's usefulness. Concerns also arose about the lag time in service provision, particularly regarding reactivation if enrollment expired, with waiting periods lasting 1–4 months.

**Patient load.** All the key informants reported a substantial increase in patient visit ranging from approximately 20% to 70% to their health facilities after adopting the insurance program. Some PHCCs reported almost half of their total patients were insured whereas private facilities reported that nearly one-fourth of their patient population consisted of insurance enrollees. The NHIP's requirement to use primary service points was identified as a major factor contributing to the increased patient load in public health facilities. A few PHCCs even reported that they were mostly seeing referral cases.

As many focus group participants complained of overcrowding, and Key Informants reported a substantial increase in patient visits after adopting NHIP, we examined the treatment effect of policy on patient visits between intervention and comparator districts using the DID method. The findings (Table 3) suggested an increase in the average number of total

**Table 3.  Difference in difference estimation.**

| Variable | Interaction co-eff | 95% CI | | p-value |
|---|---|---|---|---|
| **All health facilities** | | | | |
| Total client visits | 4729 | −3801 | 13259 | 0.28 |
| New client visits | 2721 | −4207 | 9649 | 0.44 |
| Total referral outs | 163 | 58 | 268 | 0.01* |
| **Sub-group analysis** | | | | |
| *Hospitals* | | | | |
| Total client visits | 10072 | −20527 | 40671 | 0.52 |
| New client visits | 5065 | −20363 | 30493 | 0.69 |
| Total referral outs | 39 | −212 | 290 | 0.76 |
| *PHCCs* | | | | |
| Total client visits | 3051 | −731 | 6833 | 0.1** |
| New client visits | 2031 | −306 | 4368 | 0.09** |
| Total referral outs | 192 | 88 | 296 | <0.001* |

$H_0$: the means are not different. *P < 0.05: reject null hypothesis at 5%. **P < 0.1: reject null hypothesis at 10%.

and new client visits by 4,729 and 2,721, respectively, in overall health facilities of intervention districts compared to comparator districts, due to the policy's effect. Similarly, the average number of referral outs increased by 163 in intervention districts. However, referral outs were only statistically significant for PHCCs. In subgroup analysis, we found a statistically significant increase in client visits and referrals for PHCCs but not for hospitals. Nonetheless, the findings indicated an increase in client visits by an average of 9 clients per day in PHCCs and 32 clients per day in hospitals, which reaffirmed the claims from key informants.

## Discussion

This study found a complex relationship between demand and supply factors affecting insurance program enrollment and healthcare utilization. Our findings on key determinants of health insurance enrollment in Nepal align with other studies, highlighting factors such as higher socioeconomic status, privileged ethnic groups, chronic illness in the family, knowledge regarding health insurance, willingness to pay, access to health facilities, availability of medicines, health worker behavior, and service lag time [35,36]. A study in India revealed a higher likelihood of enrollment among female-headed households [37]. Although Nepali society, like India, is patriarchal with male-dominated households, our female participants reported mutual understanding with male family members regarding insurance. Marginalized groups might have been overlooked during enrollment, as EAs seemed to enroll those who approached them. Higher-income individuals might be employed, and unless formal sector enrollment is mandatory, these populations could be missed. The perceived usefulness of the NHIP was also a key determinant for enrollment.

Our quantitative analysis reiterated the claim of increased patient flow and referral outs in health facilities, especially the PHCCs. However, the increase in client visits at hospitals did not reach statistical significance, likely due to the small hospital sample size, short NHIP implementation period, geographic infeasibility of district hospitals for many enrollees, and distrust in district hospitals as referral centers, leading enrollees to private or tertiary care centers. The quality of health services were often influenced by various NHIP design features and implementation issues like claim difficulties, delayed reimbursements, cumbersome medicine procurement, and the requirement of primary service points, among others.

NHIP's design closely resembles the Social Health Insurance Systems model by Providing for Health (P4H), a partnership between France, Germany, ILO, WHO, and the World Bank [38]. However, context and implementation processes vary among countries, leading to different outcomes in universal health coverage. Regardless, enrollment in NHIP primarily depends on the quality of healthcare delivery system. Increasing sanctioned staff positions and the availability of essential medicines under free health services are some examples that play crucial roles in maintaining the quality of healthcare system. This requires substantial planning and coordination between HIB and the Ministry of Health and Population (MOHP).

Our findings also suggest the following implications.

Adverse selection: Despite policy changes, voluntary enrollment continues in NHIP. A study found that approximately one-third of the current enrollees receive subsidies [17], consistent with our qualitative assessment indicating higher enrollment of older individuals and dropout of healthier ones. While Family-based membership helps [39], our post-focus group assessment revealed that a quarter of households only partially participate, often enrolling high-risk or chronically ill members. Though our focus group sampling may lack causal inferences, comparable instances of partial enrollment exist in other countries [40], and several studies indicate chronic illness as a predictor for insurance enrollment in Nepal [18,19].

Enforcing mandatory enrollment could address adverse effects, but it is challenging. The Philippines achieved high population coverage despite voluntary enrollment, but nearly half of those enrollees were subsidized, raising sustainability concerns [39,41]. Nepal is pursuing a similar strategy with free and subsidized premiums for certain groups, albeit without revenue streams like the Philippines' "sin tax" [41]. Issues such as false wage reporting, evasion of registration, and employer non-compliance can arise even in the formal sector enrollment [42]. Korea achieved universal coverage with mandatory enrollment, mostly due to its higher economic growth [42]. However, Nepal's fluctuating GDP growth [43] makes this approach challenging. Alternatively, Thailand's tax-based system presents an option but requires steadfast political commitment and a robust healthcare delivery system, particularly in disadvantaged areas [44], which is also a challenge for Nepal.

*Financial risk protection*: One of the key objectives of social health insurance entails providing financial risk protection against catastrophic health expenditures, which necessitates balancing a comprehensive benefits package, coverage of the poor, and financial sustainability [5,45]. A preliminary estimate from Columbia suggests that achieving this may cost close to 20 percent of the GDP [45], a figure beyond Nepal's economic capacity. Currently, NHIP offers modest benefits, with a benefit ceiling of NPR 100,000 for a family of five members, and faces challenges such as non-acceptance by many non-profit service providers.

Despite its gate-keeper function, our qualitative findings suggest that many individuals primarily use primary-care facilities for referrals, often opting for private hospitals. While referral to private facilities might help with access to care, one needs to note that private facilities are charging the gap rate to the insurance enrollees and NHIP lacks a clear statement prohibiting this practice. This could inflate the OOP spending, mirroring the situation in the Philippines where insured and uninsured patients faced comparable costs due to increased price-cost margins to insured patients [46]. Moreover, many insurance enrollees were reluctant to use public facilities for severe illness due to wait times and service quality concerns, diminishing the perceived value of the insurance program.

*Fund shifting from public to private entity*: Gertler and Strum estimated a 33 percent reduction in public delivered care in Jamaica after expanding SHI [47]. We do not have such an estimate for Nepal, but the annual report from HIB indicates a large portion of reimbursement goes to the private facilities [16]. It signifies the shifting of public funds to the private entity even though SHI's objective is to reduce financial pressure on public budgets [48].

*Inequitable access to health care*: Nepal is trying to provide equitable access to care by subsidizing premiums to poor and other disadvantaged groups. However, it has not been effective due to insufficient means-testing and health facilities' inability to cater to needs. Besides, the insurance program is expanded only up to the PHCC level, and most of the referral health facilities are centered around the cities. This raises a concern about equity for the rural population and portends that the insurance program might transfer funds from rural disadvantaged people to more affluent urban populations.

*Increase in health care utilization*: Both qualitative and quantitative findings suggest the increase in overall patient flow in health facilities with the health insurance program. Our quantitative analysis reaffirmed claims of increased patient flow in health facilities, revealing an average increase of 9 patient visits per day in PHCCs and 32 patient visits per day in hospitals. This aligns with a separate study indicating a utilization rate of 88.7% [20]. Notably, this increase was observed when the enrollment rate to the insurance program was only around 5% [13–16]. The lack of concurrent increase in sanctioned posts for health service providers indicates the need for additional resources; without which the quality of care may further deteriorate as enrollment rate grows.

*Strengths of the study*: The study's design provided rich insights into participants' experiences. While our sampling is purposive, we incorporated perspectives from diverse participants across various geographic regions, reaching data saturation. Furthermore, our findings are consistent with other studies both within and outside Nepal. We also provided a preliminary assessment of health service utilization using national health system data (DHIS-2), which is a novel approach.

*Limitations of the study*: While this study explored the experiences of beneficiaries and providers, it did not include the perspective of the purchaser (i.e., Health Insurance Board). Many tertiary-level health facilities either do not participate in or withdraw from the insurance program, requiring in-depth exploration. Our quantitative analysis findings should be treated as preliminary. Due to missing data in DHIS-2, we couldn't account for facility-level confounders such as the availability of resources. Notably, these covariates may serve as mediators rather than confounders, given that NHIP's objective is to enhance health care quality. Data from private and tertiary referral centers were lacking, as they are not mandated to report to DHIS-2, potentially underestimating the NHIP's overall impact. Our study also lacks an estimate of adverse selection, financial protection, or equitable access to care, necessitating further investigation. The post-focus group survey had a higher proportion of respondents who were female, Dalit, had higher annual incomes, and used private health facilities explicitly among the non-enrollee group. Due to the purposive sampling, we cannot claim it is a true representation of the general population and requires further investigation.

## Conclusion

This study highlights the complex relationship between demand and supply factors affecting health insurance enrollment and subsequent health service utilization. Perceptions of service quality, including resource availability, play a crucial role in enrollment and renewal decisions. Service provision challenges and provider behavior contribute to beneficiary dissatisfaction, necessitating improved service delivery and provider training. Accessibility issues, long waits, and referral complexities drive individuals to private healthcare and out-of-pocket payments, undermining the insurance program's value. Lack of awareness and financial concerns hinder enrollment efforts, emphasizing the need for targeted outreach and community engagement. Despite these challenges, healthcare utilization has increased significantly post-implementation. Ensuring quality healthcare under the insurance program is vital for retaining enrollees and motivating non-enrollees to join. Addressing these challenges requires a holistic approach, including improved coordination between MOHP and HIB, policy reforms, community engagement, and enhanced service delivery to ensure equitable access to quality healthcare.

## Supporting information

**S1 Table. List of Intervention and comparator districts.**
(DOCX)

**S1 Text. Data collection instruments.**
(DOCX)

**S1 Fig. Graphical presentation of trends for client visits and referral outs from Primary Health Care Centers and Hospitals.**
(TIF)

## Acknowledgments

We extend our gratitude to the CUNY Graduate School of Public Health and Health Policy, the Nepal Health Insurance Board, including its province and district offices in Kaski, Baglung, and Kailali, as well as the Integrated Health Management Section of the Department of Health Services Nepal. Special thanks go to the Enrollment Assistants, health service providers, and the community people from Kailali and Baglung Districts of Nepal.

## Author contributions

**Conceptualization:** Rajani Bharati, Diana Romero, Alexis Pozen, James Sherry.

**Data curation:** Rajani Bharati, Mukesh Adhikari.

**Formal analysis:** Rajani Bharati.

**Funding acquisition:** Rajani Bharati.

**Investigation:** Rajani Bharati, Diana Romero, Alexis Pozen, Mukesh Adhikari, Prakash Acharya.

**Methodology:** Rajani Bharati, Diana Romero, Alexis Pozen, James Sherry, Bhuwan Paudel, Mukesh Adhikari.

**Project administration:** Rajani Bharati, Bhuwan Paudel, Prakash Acharya.

**Resources:** Bhuwan Paudel, Prakash Acharya.

**Software:** Rajani Bharati.

**Supervision:** Diana Romero, Alexis Pozen, James Sherry, Bhuwan Paudel.

**Validation:** Alexis Pozen, Prakash Acharya.

**Visualization:** Prakash Acharya.

**Writing – original draft:** Rajani Bharati.

**Writing – review & editing:** Rajani Bharati, Diana Romero, Alexis Pozen, James Sherry, Bhuwan Paudel, Mukesh Adhikari, Prakash Acharya.

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
