## [Decision Letter · Decision Letter 0]

26 Jun 2024

PGPH-D-24-00962

Experiences and effect of implementing Social Health Insurance (SHI) program in Nepal- A mixed method study.

Dear Dr. Bharati,

Thank you for submitting your manuscript to PLOS Global Public Health. After careful consideration, we feel that it has merit but does not fully meet PLOS Global Public Health’s publication criteria as it currently stands. Therefore, we invite you to submit a revised version of the manuscript that addresses the points raised during the review process.

We look forward to receiving your revised manuscript.

Kind regards,

Carl Abelardo T. Antonio

Academic Editor

Journal Requirements:

Additional Editor Comments (if provided):

This manuscript requires further clarification before it can be considered for publication. Please refer to the reviewer comments

Reviewers' comments:

Reviewer's Responses to Questions

**Comments to the Author**

1. Does this manuscript meet PLOS Global Public Health’s publication criteria ? Is the manuscript technically sound, and do the data support the conclusions? The manuscript must describe methodologically and ethically rigorous research with conclusions that are appropriately drawn based on the data presented.

Reviewer #1: Yes

Reviewer #2: Yes

2. Has the statistical analysis been performed appropriately and rigorously?

Reviewer #1: N/A

Reviewer #2: Yes

3. Have the authors made all data underlying the findings in their manuscript fully available (please refer to the Data Availability Statement at the start of the manuscript PDF file)?

Reviewer #1: No

Reviewer #2: Yes

4. Is the manuscript presented in an intelligible fashion and written in standard English?

Reviewer #1: Yes

Reviewer #2: Yes

5. Review Comments to the Author

Reviewer #1: This is an important research area that requires further national and global attention. I have provided some points as suggestions to refine the manuscript and report of findings. Please also see my notes as comment bubbles in attached PDF.

Introduction

•After reading this, I’m still left wondering about the history of policies, and what is included, how is it delivered, and why there is a need for enrolment as opposed to automatic universal enrolment. More details and clarity of the process will be helpful.

•Why is public financing alone not feasible in Nepal? Please add a line or two on this to hook onto global health discussions on this topic.

•The introduction section will benefit from more discussions of published international studies on the pros and cons of social insurance systems, and why it is the policy option chosen in national circumstances, and what are known challenges and enablers in other low and middle income countries.

•Useful papers that could be considered: https://www.ncbi.nlm.nih.gov/pmc/articles/PMC9648174/

•https://www.thelancet.com/journals/langlo/article/PIIS2214-109X(23)00510-7/fulltext

•https://bmcpublichealth.biomedcentral.com/articles/10.1186/s12889-020-08742-1

•https://www.ncbi.nlm.nih.gov/pmc/articles/PMC10428604/

Methods

•To clarify- was on overall grounded theory method used for the study including identification of participants? Or just in analysis? Please clearly describe this, and why/how did the grounded theory analysis be considered suitable, and what were the novel theories or themes identified from this process? How did the analysis method differ from thematic analysis etc?

Results and Discussions

•On the demand side, how did demographic and intersectionality affect enrolment, use of services and drop out? Why? You’ve provided the categories represented in research, but not so much details of how these categories influenced demand-related perceptions and decisions.

•The fishbone diagram is useful to visually portray the themes and anchor report and discussions.

•How were themes distinctly related to the design and implementation of social insurance program as opposed to broader health system factors- for instance- you mention quality as one of the biggest consideration, but how is that tied to social insurance design and implementation, as opposed to broader health system factors? Related procurement policies, and service quality. Please describe and discuss.

•Perhaps it is outside the scope of the results but worthy of a discussion or identification of future research= how are demand related factors influenced by age, gender, education, employment status and income levels, etc? Who precisely are benefitting, who are not and dropping out? For instance- gender and social insurance or employment related health entitlements is an area that can influence demand or rather even when there is demand, the ability of continuously access timely and necessary health care- see these publications for context:

•https://www.unwomen.org/sites/default/files/Headquarters/Attachments/Sections/Library/Publications/2020/Discussion-paper-Universal-health-coverage-gender-equality-and-social-protection-en.pdf

•https://bmchealthservres.biomedcentral.com/articles/10.1186/s12913-023-10473-z

•https://www.ncbi.nlm.nih.gov/pmc/articles/PMC5886176/

•https://www.bmj.com/content/371/bmj.m3384

•https://p4h.world/en/is-health-financing-gender-biased/

Reviewer #2: PLOS GLOBAL PUBLIC HEALTH

REVIEWERS COMMENTS

PGPH-D-24-00962: Experiences and effect of implementing Social Health Insurance (SHI) program in Nepal-A mixed method study

Dear Editor,

I submit to you my comments on the above manuscript submitted to PGPH for review

General comments

Overall, the intentions of the manuscript and the issues discussed are relevant. However, the authors need to place the results in perspective to enhance its understanding.

Specific comments

I provide some specific comments here, which will include comments and questions.

1.The presentation of the results section of the abstract is very general. Per the research objectives, the authors should present the results accordingly i.e. providers, client and then the effect.

2.Again in the abstract, there was no results from the DID analysis for the effect of enrolment on utilization.

3.Abstract: the conclusion seems not to emanate from the findings of the research. There were specific client and provider results to which the study can conclude on as well as make recommendations. The current conclusion is focusing on collaboration between the Health Insurance Board and the Ministry. These are important but they do not the direct results from the study or the objectives to which the study was set.

4.Study objectives: Lines 133-138. I suggest that the objectives are not numbered. They could be considered as a small paragraph at the end of the introduction

5.‘FY’…please write in full for the first time it appears in the manuscript

6.Methods: What informed the use of the exploratory sequential mixed methods design? In my simple understanding of the sequence it implies qual-quant-qual sequence. However, per the data collection, analysis and results presented, it seems like the first two objectives followed the qualitative approach while the last objective followed the quantitative approach.

7.It does not look like the qualitative informed the quantitative approach, and neither did the quantitative approach added information to the qualitative. Per the way the results have been presented. That linkage is missing

8. From lines 244 in the results section, the themes were just presented. The reader is not lead to know what exactly is going to be discussed. Do they relate to the first objective? Similarly as you read along, then the findings that relate to the providers experience also follow but the reader is not guided. For instance I will expect that before for example line 244, you introduce briefly the major themes that emanated from the clients experience and then the details will follow. This should be the same for the providers

9.I also did not see any connection between the qualitative findings and the quantitative findings. If there are no connections or the studies were conducted independently, then this should be stated clearly and a revision of the type of mixed methods study design

10.Lines 270-272: the quote relates to human resource skills than availability of medicines and diagnostic facilities per the theme.

11.Major findings and key issues in the study did not come out clearly. I find some of the issues addressed in the discussion not coming from the results-adverse selection, financial risk protection for example are two such issue that was not clearly indicated in the results.

Thank you

6. PLOS authors have the option to publish the peer review history of their article (what does this mean? ). If published, this will include your full peer review and any attached files.

**Do you want your identity to be public for this peer review?** For information about this choice, including consent withdrawal, please see our Privacy Policy .

Reviewer #1: **Yes: ** Lavanya Vijayasingham

Reviewer #2: No

---

## [Decision Letter · Decision Letter 1]

21 Mar 2025

Experiences and effect of implementing Social Health Insurance (SHI) program in Nepal- A mixed method study.

PGPH-D-24-00962R1

Dear Dr Bharati,

We are pleased to inform you that your manuscript 'Experiences and effect of implementing Social Health Insurance (SHI) program in Nepal- A mixed method study.' has been provisionally accepted for publication in PLOS Global Public Health.

Best regards,

Julia Robinson

Executive Editor

Reviewer Comments (if any, and for reference):

Reviewer's Responses to Questions

**Comments to the Author**

1. If the authors have adequately addressed your comments raised in a previous round of review and you feel that this manuscript is now acceptable for publication, you may indicate that here to bypass the “Comments to the Author” section, enter your conflict of interest statement in the “Confidential to Editor” section, and submit your "Accept" recommendation.

Reviewer #3: All comments have been addressed

2. Does this manuscript meet PLOS Global Public Health’s publication criteria ? Is the manuscript technically sound, and do the data support the conclusions? The manuscript must describe methodologically and ethically rigorous research with conclusions that are appropriately drawn based on the data presented.

Reviewer #3: Yes

3. Has the statistical analysis been performed appropriately and rigorously?

Reviewer #3: Yes

4. Have the authors made all data underlying the findings in their manuscript fully available (please refer to the Data Availability Statement at the start of the manuscript PDF file)?

Reviewer #3: Yes

5. Is the manuscript presented in an intelligible fashion and written in standard English?

Reviewer #3: Yes

6. Review Comments to the Author

Reviewer #3: (No Response)

7. PLOS authors have the option to publish the peer review history of their article (what does this mean? ). If published, this will include your full peer review and any attached files.

**Do you want your identity to be public for this peer review?** For information about this choice, including consent withdrawal, please see our Privacy Policy .

Reviewer #3: No
